# A needs analysis of ESP courses in colleges of art and design: Consensus and divergence

**Fengfan Mao** [ID]*, **Jiefeng Zhou**

Department of Common Required Courses, Hubei Institute of Fine Arts, Wuhan, Hubei, People's Republic of China

* maofengfan@hifa.edu.cn

**Data Availability Statement:** All relevant data are within the manuscript and its Supporting information files.

**Funding:** This work was funded by the Education Research Project from Hubei Provincial

## Abstract

This study presents a comprehensive needs analysis of English for Specific Purposes (ESP) courses in colleges of art and design in China. By examining the perspectives of undergraduate students, graduate students, art teachers, and English teachers, the research identifies consensus and divergence in ESP needs and implementation challenges. The analysis reveals a strong demand among students for specialized English learning, with a particular emphasis on receptive and interactive skills. However, there is a widespread acknowledgement of students' difficulties across ESP skills, indicating a gap between student competencies and the professional demands, thus highlighting the need for targeted educational interventions to effectively address the difficulties. Notably, reading and writing abilities emerge as the most lacking skills. While there is unanimous recognition of the positive impact of ESP on students' professional abilities and international competitiveness, a significant disparity exists in the enthusiasm for implementing ESP courses, paticularly with resistance from English teachers. The study highlights the critical challenges in ESP teaching, with teacher qualifications identified as the most prominent issue. Interestingly, English teachers attribute difficulties primarily to teaching materials, indicating a potential lack of readiness for ESP course delivery. A gap is also observed between teachers' evaluations of student abilities and students' self-assessment, especially among undergraduate freshmen, suggesting an overestimation of their English proficiency in professional contexts. The paper concludes with the implication that for effective ESP course implementation in colleges of art and design in China, targeted faculty development programs and a collaborative approach involving English teachers, art teachers, and professionals are essential. The collaboration should aim to develop materials that integrate specific professional knowledge with linguistic expertise. Additionally, a balanced approach combining general English proficiency and specialized English training is recommended to address both foundational and specialized language skills. Overall, the study underscores the need for targeted ESP courses in art and design education, tailored to bridge the gap between current proficiency levels and professional language requirements.

Department of Education(Grant No. 2022401) and Education Research Project from Hubei Institute of Fine Arts. The funders had no role in study design, data collection and analysis, decision to publish, or preparation of the manuscript.

## 1. Introduction

As the global economy becomes increasingly interconnected and higher education shifts towards a more international orientation, there is an escalating demand for professionals who are not only experts in their field but also proficient in English. This trend is propelled by the growing need for specialized English skills across various professional domains. Consequently, English for Specific Purposes(ESP), a branch of English language teaching within applied linguistics [1, 2], has gained significant traction. ESP is particularly focused on equipping students with the linguistic competences specific to their chosen professions, thereby enabling them to communicate effectively and excel globally in their respective fields [3–6]. In China, the trajectory of college English teaching has been progressively shifting from English for General Purposes(EGP) to ESP [7–10]. This shift reflects a broader trend observed in many countries where ESP is increasingly integrated into higher education curricula [11–14].

The most recent edition of *Handbook for College English Teaching* [15], a pivotal guideline for college English curriculum design in China, advises universities to develop distinctive English courses tailored to their specific needs. These courses are categorized into three groups: EGP, ESP, and cross-cultural communication. Within ESP, further subdivisions include English for Academic Purposes(EAP) and English for Occupational Purposes(EOP) [15]. EAP courses have seen considerable development in Chinese universities, as evidenced by the proliferation of textbooks on academic reading and writing, comparatively matured source of instructors [16], increased research in EAP [17], and the availability of online courses on major platforms such as UMOOCS and icourse163 [18, 19]. However, ESP courses are still in their nascent stages, with uneven distribution across disciplines. This variability is attributed to diverse professional backgrounds and differing perceptions of ESP [20]. While mature ESP courses in China align with international trends, encompassing areas like English for Science and Technology, Business English, Legal English, Medical English, and Journalism English [18, 21], the field of art and design has been relatively under-explored [22].

Against this backdrop, this study employs a needs analysis approach to identify specific requirements for ESP courses and the perceptions of relevant stakeholders that may influence their development in the domain of art and design. By utilizing a questionnaire, the research aims to uncover insights that can inform ESP course design in colleges specializing in art and design.

## 2. Literature review

Since its inception, ESP has been fundamentally oriented towards addressing the specific needs of learners. The concept of needs analysis is central to ESP, as it ensures the relevance and effectiveness of the course content [23]. Hutchinson and Waters and Harding emphasize that needs analysis is crucial in every ESP course design, typically marking the starting point in curriculum development [6, 24]. Needs analysis encompasses the learners' goals, current language proficiency, and the linguistic skills they need to develop, thereby ensuring that course content is precisely tailored to their specific requirements [25]. This review examines some of the most influential models for needs analysis in the field of ESP.

### 2.1 Munby's target-situation-oriented approach

John Munby offers a systematic approach to needs analysis in language curriculum development [26]. Needs analysis is conceptualized as a mechanism to bridge the gap between the linguistic demands of specific social contexts and the learner's capabilities. His framework emphasizes the importance of identifying target situations where language is used, characterizing these situations in terms of communicative activities, and defining the linguistic elements

required for effective communication within these contexts. Munby introduces the 'Communicative Needs Processor' (CNP) to operationalize the analysis of learners' needs by translating real-world communication demands into classroom syllabus objectives [26]. The strength of Munby's approach lies in its detailed and structured method, which ensures that all relevant aspects of communication are considered when designing a syllabus tailored to specific needs. His work has been influential in highlighting the role of authentic, context-specific language use in syllabus design, advocating for curricula that reflect the realistic use of language in its social context.

However, despite its pivotal role, the CNP has faced critiques for its intricate parameters and a predominant focus on linguistic elements at the expense of considering the practical contexts in which language is employed. Critics like West [27] and Long [28] have pointed out its potential limitations in addressing the broader, contextual demands of language use.

## 2.2 Hutchinson and Waters' learning-centered approach

While Munby's CNP was groundbreaking, it also demonstrated the limits of a strictly language-centered approach to needs analysis. Hutchinson and Waters critique this approach by arguing that a deep understanding of both target and learning needs can enhance the relevance and effectiveness of ESP courses. They advocate for a balanced consideration of what learners will do with the language and how they will learn it, proposing a more learner-centered approach to course design [6].

Hutchinson and Waters expand on the concept of needs analysis in their work, emphasizing that it should begin with understanding why learners need English. Their model distinguishes between 'target needs' (what learners need to do in the target situation) and 'learning needs' (what learners need to achieve these goals). They further differentiate target needs into '-necessities,''lacks,' and 'wants'—categories that help educators understand not only the objective requirements of the target situation but also the subjective experiences and desires of the learners. This approach is lauded for its holistic view, integrating linguistic requirements with how language acquisition occurs, and has been applied widely to analyze data [14, 29, 30].

The frameworks developed by Munby, and Hutchinson & Waters have significantly shaped the ESP field. They highlight the complexity of needs analysis and the importance of integrating both target situation requirements and learner preferences into course design. This dual focus ensures that ESP programs are both relevant to the learners' professional or academic contexts and responsive to their personal learning needs and motivations.

## 2.3 Dudley-Evans & St John's multi-disciplinary approach

Dudley-Evans and St. John introduces a multi-disciplinary approach to needs analysis in ESP, emphasizing the iterative and cyclical nature of the process [5]. This perspective adds a dynamic component to the frameworks by Hutchison & Waters and Munby, highlighting the ongoing interplay between needs analysis and course evaluation. This concept aligns with modern pedagogical practices that prioritize adaptability and responsiveness to learner feedback and changing conditions.

This needs analysis model also elaborates on the broadening definitions of needs, incorporating objective, subjective, perceived, and felt needs, which are categorized into product-oriented(target situation analysis) and process-oriented(learning situation analysis) needs. This expansion allows for a more nuanced understanding of both the learners' requirements in the target situation and their learning processes, addressing the criticism Munby faced for his somewhat rigid communicative needs processor.

Another significant contribution of this approach is the concept of 'means analysis', which focuses on the learning environment and acknowledges that the successful implementation of a course depends not just on the identified needs but also on the context in which the course operates.

This multi-disciplinary approach to needs analysis in ESP enriches the traditional frameworks by integrating continuous evaluation into the process, advocating for a broader understanding of needs that includes logistical and environmental considerations, and emphasizing the need for adaptability in course design and implementation. This approach effectively addresses the dynamic and often complex nature of language use and learning within specific professional or academic contexts.

## 2.4 Basturkmen's comprehensive approach: Bridging gaps and integrating contextual realities

Basturkmen's research offers a robust extension to the existing frameworks on needs analysis by emphasizing practical applications and integrating real-world educational challenges into the needs analysis process [31, 32]. It focuses particularly on the operationalization of needs analysis in ESP course development, highlighting the dynamic interplay between theory and practice.

This approach emphasizes the necessity of continuous needs analysis that goes beyond pre-course assessments to include ongoing evaluations throughout the course delivery. This perspective aligns well with the multi-disciplinary approach but adds specific methods for integrating findings into course design in real-time, thus ensuring that ESP courses remain relevant and responsive to both learner and contextual changes.

A variety of analytical methods are introduced to perform needs analysis, including discourse analysis, present situation analysis, and teaching context analysis. These methods are designed to provide a holistic view of learners' needs by not only focusing on the linguistic requirements but also considering the educational environment and the learners' personal and cultural backgrounds [17]. This approach significantly expands upon earlier models by detailing how each component of the analysis can be practically applied to enhance ESP course development.

Basturkmen's research enriches the existing literature on needs analysis by providing detailed, practical strategies for continuous assessment and adjustment of ESP courses. It extends the previous theoretical models by introducing a comprehensive suite of tools for analyzing and integrating the diverse needs of learners, thereby ensuring that ESP courses are both effective and adaptable.

## 2.5 Summary

Needs analysis has developed from simple teacher intuitions to sophisticated frameworks involving detailed empirical data and comprehensive theoretical underpinnings. This evolution reflects the growing complexity and specificity required in ESP to cater to diverse and changing learner needs effectively. It was initially a simple pre-course activity focusing mainly on target situation analysis. Over time, it has evolved into a sophisticated, multi-faceted procedure that encompasses the analysis of deficiencies, context, strategies and means [33]. It is not a static, one-time activity but a dynamic component of curriculum effectiveness; it should consider not only the immediate linguistic needs but also the sociopolitical, psychological and contextual conditions affecting learners. Data collection methods for conducting needs analysis involves interviews, observations, and questionnaires [21].

## 3. Theoretical framework

The theoretical framework for this research on needs analysis in colleges of art and design is a pre-course anlaysis, primarily anchored in the models provided by Basturkmen [32], as well as incorporating significant insights from Munby [26], and Hutchinson & Waters [6]. These models offer a comprehensive approach to understanding the specific linguistic, communicative, and contextual needs of learners in specialized settings.

Basturkmen [32] emphasizes a holistic approach to needs analysis that involves multiple layers of investigation: target situation analysis (TSA), present situation analysis (PSA), and teaching context analysis (TCA). These components are crucial for developing ESP courses that are both responsive to the learners' current abilities and aligned with their future professional requirements.

Target Situation Analysis (TSA): This aspect of the needs analysis focuses on identifying the specific language and communication skills that students will need in their future professional contexts. It is essential for understanding what the end goals of the ESP course should be [26].

Present Situation Analysis (PSA): PSA assesses learners' current language proficiency. It helps in identifying the gap between their present level of English and the level required for their professional needs. This analysis is vital for tailoring the ESP curriculum to address these gaps effectively.

Learner Factor Analysis (LFA): This considers the learners' attitudes, motivations, and preferences in learning English. Understanding these factors is crucial for designing an ESP course that is engaging and effective.

Teaching Context Analysis (TCA): TCA examines the educational environment, including resources available, institutional support, teacher expertise and any constraints that might impact the delivery of the ESP program. This is particularly relevant in the context of art and design education, where the integration of language skills with artistic and design concepts can be challenging [14].

The framework for this study is shown in the diagram (Fig 1). It integrates Hutchinson and Waters' [6] distinctions between learners' needs (TSA), wants (LFA), and lacks (PSA). This approach ensures that the course content is not only necessary for the learners' future roles but also aligns with their personal aspirations and perceived needs, which is particularly relevant in creative fields like art and design.

By synthesizing these models, the theoretical framework for this study advocates for a comprehensive, responsive, and flexible approach to designing ESP courses in art and design colleges.

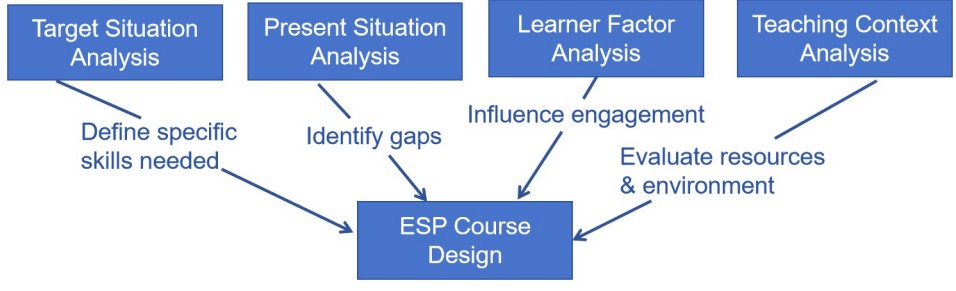

**Fig 1. Framework for analysis.**

## 4. Research design

### 4.1 Ethics statement

This research, focused on investigating the specific needs of English for Specific Purposes (ESP) courses in colleges of art and design, is designed to uphold the highest ethical standards. The study is investigation-oriented and does not involve the disclosure of any personal information that could lead to the identification of individual participants, thereby posing no risks or ethical concerns. Participants were recruited on a volunteer basis, with the recruitment period spanning from May 20th, 2023 to June 14th, 2023. In the introduction part of the survey, participants were informed about the purpose of the study and assured of the anonymity of their responses. The statement clarified that participation was entirely voluntary, and respondents could terminate the survey without negative consequences; completion and submission of the survey were considered as implicit consent for the use of the provided data in future research publications. While formal written consent was not obtained, the procedure was in line with the ethical guidelines provided by the Research Office of the author's institute, which formally confirmed on May 8th, 2023 that ethical approval could be exempted for this study.

To further ensure ethical compliance, all collected data is treated with strict confidentiality. Responses are stored securely and are accessible only to the research team. Data analysis and reporting are conducted in such a way that individual participants cannot be identified, and findings are presented in aggregate form.

This approach to ethics in research reflects a commitment to protecting the rights and well-being of participants, ensuring the integrity of the research process, and contributing valuable insights into the field of ESP in art and design education.

### 4.2 Research respondents

Recent research about ESP emphasizes the integration of data from various stakeholders, including learners, teachers, governing bodies, sponsors and employers, current practitioners, domain experts, past students, etc to address the multifaceted nature of ESP teaching needs [34]. This study recognizes the importance of gathering data from a diverse range of sources for effective needs analysis in ESP and data is collected in accordance with the principles outlined by Dudley-Evans & St John [5] and Long [28], which mentioned that the main source of information for needs analysis are: (1) learners; (2)teachers and applied linguists; (3) professional experts.

This study investigates four distinct groups of respondents—art teachers, college English teachers, undergraduate freshmen, and graduate students, all from two colleges of art and design. As professional experts and practitioners, art teachers provide insights into the professional needs and expectations within the field of art and design. Their perspectives are crucial for understanding the specific language requirements and industry standards. College English teachers offer a deep understanding of English teaching methodologies, and are adept at analyzing linguistic features relevant to the target situation. They also provide valuable information on students' proficiency levels and learning motivation, playing a key role as implementers of ESP teaching. The surveyed undergraduate freshmen come from various majors, including oil painting, watercolor painting, printmaking, Chinese painting, mural painting, ceramics, visual communication design, environmental art design, fashion design, fine arts, and animation, with approximately 20 students from each major. They offer a fresh perspective. Their diverse expectations for college English teaching, coupled with their lack of prior ESP learning experience, provide a unique viewpoint on initial language needs, motivation and preferences. The surveyed graduate students come from different majors mentioned

above. With prior experience in ESP learning, graduate students can offer insights into the effectiveness of current ESP courses and suggest areas for improvement. Their feedback is instrumental in identifying gaps between the target needs and the learners' present situation.

By investigating the attitudes and opinions of these four different groups towards ESP teaching, the study aims to better identify the discrepancies between target needs, learners' present situation and learning environments. This comprehensive approach will contribute to making the ESP curriculum at art schools more targeted and effective.

## 4.3 Research questions

The specific research questions of this study are designed to thoroughly investigate the needs and challenges in ESP education in colleges of art and design, guided by Basturkmen's comprehensive approach and informed by recent findings in ESP needs analysis. The research questions are as follows: (1) What are the specific target needs for English learning among students in colleges of art and design? (Target Situation Analysis) This question aims to identify the precise language skills and competence that art and design students require in their professional futures. (2) How proficient are art students in using English to address professional problems? (Present Situation Analysis)This question assesses the current level of English proficiency among art students, focusing on their ability to use English in professional contexts. (3) What are the perspectives of art teachers and students regarding the implementation of ESP teaching?(Learning Situation Analysis) This question explores the attitudes and opinions of both teachers and students towards ESP teaching, providing insights into their expectations and reservations. (4) What are the primary challenges and bottlenecks in ESP teaching within art schools? (Teaching Context Analysis) This question aims to identify the key obstacles and difficulties faced in the implementation of ESP courses, focusing on factors such as curriculum design, resource availability, and teacher preparedness. (5) What similarities and differences exist in the opinions of art teachers, English teachers, undergraduate students, and graduate students regarding ESP teaching? This question seeks to compare and contrast the views of different stakeholder groups, highlighting areas of consensus and divergence.

By addressing these research questions, the study aims to provide a comprehensive understanding of the needs and challenges in ESP teaching in art and design education, contributing to the development of more effective and targeted ESP courses.

## 4.4 Research methods

The research methodology adopted for this study involves a questionnaire survey, inspired by the framework designed by Cai Jigang and Chen Ningyang for surveying ESP education needs in research-oriented universities in Shanghai [35].

The questionnaire consists of 23 questions (see S1 File), covering five key aspects: (1) Identification of respondents. Question 1 asks respondents to identify their group: undergraduate freshmen, graduate students, art teachers or English teachers. This categorization is crucial for analyzing data based on different perspectives. (2) Art professional development needs. Questions 2–6 explore the specific needs of ESP for art professional development. This section is designed to understand the target situation needs of students and teachers in the art and design context. (3) Current ESP competence: Questions 7–11 survey students' present ESP competence. This part assesses the present situation analysis, focusing on the existing English skills of the students. (4) Opinions on ESP teaching: Questions 12–19 investigate the opinions of four different groups on ESP teaching. These questions aim to gather diverse perspectives on the learning situation analysis, including attitudes and motivations towards ESP. (5) Teaching situation and challenges: Questions 20–23 investigate the current teaching situation and main

**Table 1. Reliability analysis of the ESP needs survey.**

| Question numbers | Research topics | Number of items | Cronbach's alpha |
|---|---|---|---|
| 1–23 | Needs analysis of ESP courses | 23 | 0.918 |
| 2–6 | Art professional development needs | 5 | 0.829 |
| 7–11 | Current ESP competence | 5 | 0.87 |
| 12–19 | Opinions on ESP teaching | 8 | 0.847 |
| 20–23 | Teaching situation and challenges | 4 | 0.788 |

problems in ESP teaching. This section is aligned with the teaching context analysis, identifying challenges and bottlenecks in the current ESP teaching framework.

The questionnaire was reviewed by two experts prior to the data collection. Both had at least seven years of ESP teaching experience and 15years of EGP teaching experience in colleges of art and design. They were provided with a copy of the questionnaire to review all questions, make changes, and add or delete relevant items. The questionnaire was originally in Chinese for better clarity and understanding because all respondents are Chinese. For improved readability, a bilingual edition of the questionnaire is provided in the S1 File.

## 4.5 Data collection and analysis

This survey successfully collected a total of 515 valid questionnaires, comprising 227 from undergraduate freshmen(UF), 226 from graduate students(GS), 28 from art teachers(AT), and 34 from college English teachers(ET). The questionnaire utilizes a Likert 5-point scale (5 = strongly agree; 4 = agree; 3 = uncertain; 2 = disagree; 1 = strongly disagree) to gauge respondents' attitudes and perceptions. The original collected data is available in the S1 Table, ensuring transparency and facilitating further analysis or replication of the study's findings.

For data analysis, the study employed SPSS statistical software. The proportions and means of each option for each question were calculated. The analysis involved counting the number of respondents who generally agreed or disagreed, excluding those who were uncertain. Given the non-normal distribution of the results, non-parametric tests were used to examine the statistical differences between groups. When differences were not apparent, t-tests were conducted for pairwise group comparison to glean more detailed data. Additionally, paired t-tests were utilized to analyze the differences between questions.

After the data collection phase, Cronbach's coefficients were calculated to ensure the instrument's internal reliability. The instrument had an internal reliability coefficient of 0.918, and the reliability for the four aspects were satisfactory, as indicated in the following table (Table 1).

## 5. Analysis and findings

### 5.1 Art professional development needs

Table 2 reveals consistent recognition of the importance of ESP skills among the four groups—undergraduates, graduate students, art teachers, and English teachers—in professional development contexts in art and design education. Notably, there is a high level of consensus on the necessity of developing listening and reading skills, with average agreement rates of 87.30% and 86.95% respectively. These findings underscore a generally shared view on the essential nature of these skills across various professional roles within the field of art studies.

Consensus on Receptive Skills: The uniform agreement on listening and reading indicates a widespread acknowledgment of these competencies as foundational for professional success in

**Table 2. Art professional development needs.**

| Questions | Respondents | 5 (%) | 4 (%) | 3 (%) | 2 (%) | 1 (%) | Average | Generally agree (%) | Generally disagree (%) | Non-parametric test | |
|---|---|---|---|---|---|---|---|---|---|---|---|
| | | | | | | | | | | H-value | P-value |
| 2.To enhance professional competence or meet work requirements, it is necessary to read English works, literature, and articles in the field of art and design. | UF | 57.08 | 34.51 | 7.08 | 0.44 | 0.88 | 4.46 | 91.59 | 1.32 | 7.198 | 0.066 |
| | GS | 55.31 | 37.17 | 6.19 | 0.88 | 0.44 | 4.46 | 92.48 | 1.32 | | |
| | AT | 50 | 35.71 | 7.14 | 3.57 | 3.57 | 4.25 | 85.71 | 7.14 | | |
| | ET | 55.88 | 23.53 | 17.65 | 0 | 2.94 | 4.29 | 79.41 | 2.94 | | |
| | average | 54.57 | 32.73 | 9.52 | 1.22 | 1.96 | 4.37 | 87.30 | 3.18 | | |
| 3.To enhance professional competence or meet work requirements, it is necessary to attend courses and lectures delivered by foreign scholars and experts. | UF | 49.12 | 39.82 | 9.73 | 0.44 | 0.88 | 4.36 | 88.94 | 1.32 | 4.288 | 0.232 |
| | GS | 47.35 | 36.28 | 15.49 | 0.88 | 0 | 4.3 | 83.63 | 0.88 | | |
| | AT | 50 | 42.86 | 3.57 | 3.57 | 0 | 4.39 | 92.86 | 3.57 | | |
| | ET | 41.18 | 41.18 | 5.88 | 11.76 | 0 | 4.12 | 82.36 | 11.76 | | |
| | average | 46.91 | 40.04 | 8.67 | 4.16 | 0.22 | 4.29 | 86.95 | 4.38 | | |
| 4.In work or study, it is necessary to write down creative thinking, work descriptions, reports, papers, etc. in English. | UF | 35.84 | 34.51 | 25.22 | 2.65 | 1.77 | 4 | 70.35 | 4.42 | 7.254 | 0.064 |
| | GS | 33.63 | 31.86 | 27.88 | 4.87 | 1.77 | 3.91 | 65.49 | 6.64 | | |
| | AT | 14.29 | 32.14 | 46.43 | 3.57 | 3.57 | 3.5 | 46.43 | 7.14 | | |
| | ET | 32.35 | 44.12 | 11.76 | 11.76 | 0 | 3.97 | 76.47 | 11.76 | | |
| | average | 29.03 | 35.66 | 27.82 | 5.71 | 1.78 | 3.85 | 64.69 | 7.49 | | |
| 5. It is necessary to engage in oral communication or presentations related to art profession in English, or to present papers in English at academic conferences. | UF | 38.50 | 38.50 | 19.47 | 2.21 | 1.33 | 4.11 | 77 | 3.54 | 10.197 | 0.017* |
| | GS | 31.86 | 31.86 | 30.09 | 5.31 | 0.88 | 3.88 | 63.72 | 6.19 | | |
| | AT | 35.71 | 42.86 | 10.71 | 7.14 | 3.57 | 4 | 78.57 | 10.71 | | |
| | ET | 32.35 | 38.24 | 14.71 | 2.94 | 11.76 | 3.76 | 70.59 | 14.7 | | |
| | average | 34.61 | 37.87 | 18.75 | 4.40 | 4.39 | 3.94 | 72.47 | 8.79 | | |
| 6.There is a demand for study tours, further education abroad, or visiting scholars. | UF | 30.09 | 28.76 | 33.63 | 6.19 | 1.33 | 3.8 | 58.85 | 7.52 | 24.118 | 0.000** |
| | GS | 28.76 | 27.88 | 33.19 | 7.52 | 2.65 | 3.73 | 56.64 | 10.17 | | |
| | AT | 64.29 | 32.14 | 3.57 | 0.00 | 0.00 | 4.61 | 96.43 | 0 | | |
| | ET | 55.88 | 29.41 | 8.82 | 0.00 | 5.88 | 4.29 | 85.29 | 5.88 | | |
| | average | 44.76 | 29.55 | 19.80 | 3.43 | 2.47 | 4.11 | 74.30 | 5.89 | | |

When * p<0.05 ** p<0.01, the difference is statistically significant, the same applies for subsequent cases.

the arts. Despite the absence of statistically significant differences across groups (p-values were above the 0.05 threshold for these questions), the high agreement rates suggest a universal curriculum focus that these areas continue to demand within art and design education.

Divergence in International Exposure: A significant finding emerges from Question 6 regarding the demand for study tours and further education abroad. The analysis shows a high H-value of 24.118 with a p-value of 0.000**, highlighting significant disparities among the groups. Art teachers and English teachers particularly value these opportunities, viewing them as crucial for professional growth and international exposure. In contrast, students, particularly undergraduates and graduates, exhibit considerably lower enthusiasm. This divergence might reflect varying perspectives on the immediate relevance or financial feasibility of international experiences for students.

Productive Skills: The overall agreement rate for the necessity of written skills is the lowest at 64.69%, with no significant difference between the four group, indicating least perceived demand for this skill. The overall agreement rate for the necessity of oral communication skills in English was 72.47%. However, postgraduate students displayed a notably lower level of agreement (p = 0.017*) for oral communication skills, suggesting distinct needs or perhaps a

different evaluation of the importance of these skills in their professional development. This variance could be attributed to the specific academic and career stages of postgraduate students, who might prioritize reading over oral skills. The distinct needs of postgraduate students regarding oral communication point towards a potential need for curriculum adjustments or additional support tailored to their specific professional trajectories.

The analysis in Table 3 utilizes paired t-tests to compare the mean scores representing perceived necessity for various ESP skills: reading, listening, writing, oral communication, and studying abroad. These tests are crucial in determining significant differences between pairs of skills, highlighting how respondents prioritize different aspects of their professional development.

Receptive vs. Productive Skills: A significant difference was observed in Pair1(T = -4.414, $p < 0.001^{**}$), with reading English literature and articles being rated as slightly less urgent than attending courses and lectures by foreign scholars. This suggests that while both skills are valued, there might be a slightly greater immediacy or applicability perceived in direct interaction with experts through courses. A substantial difference in Pair2 (T = -10.812, $p < 0.001^{**}$) indicates that attending courses is considered significantly more important than writing in English. This result may reflect a stronger emphasis on gaining knowledge and insights from external experts than on producing written content, possibly due to the immediate benefits of interactive learning experiences.

Comparative Analysis of Productive Skills: No significant difference was found in Pair3 and Pair4, suggesting a relative parity in the perceived importance of writing and speaking, and a comparative valuation between producing written content and gaining international experience. A significant difference in Pair5 (T = -2.855, $p = 0.004^{**}$) highlights that studying abroad is viewed as more crucial than engaging in oral communication. This result underscores the high value placed on international exposure and experiences over local communication skills, possibly due to the unique opportunities and broader perspective gained through such experiences.

**Table 3. Comparison of art professional development needs.**

| Pairs | Questions | Average score | Standard deviation | Mean difference | T-value | P-value |
|---|---|---|---|---|---|---|
| Pair 1 | 2.To enhance professional competence or meet work requirements, it is necessary to read English works, literature, and articles in the field of art and design. | 1.56 | 0.75 | -0.12 | -4.414 | 0.000** |
| | 3.To enhance professional competence or meet work requirements, it is necessary to attend courses and lectures delivered by foreign scholars and experts. | 1.68 | 0.77 | | | |
| Pair 2 | 3.To enhance professional competence or meet work requirements, it is necessary to attend courses and lectures delivered by foreign scholars and experts. | 1.68 | 0.77 | -0.39 | -10.812 | 0.000** |
| | 4. In work or study, it is necessary to write down creative thinking, work descriptions, reports, papers, etc. in English. | 2.07 | 0.96 | | | |
| Pair 3 | 4. In work or study, it is necessary to write down creative thinking, work descriptions, reports, papers, etc. in English. | 2.07 | 0.96 | 0.05 | 1.667 | 0.096 |
| | 5. It is necessary to engage in oral communication or presentations related to art profession in English, or to present papers in English at academic conferences. | 2.02 | 0.96 | | | |
| Pair 4 | 4. In work or study, it is necessary to write down creative thinking, work descriptions, reports, papers, etc. in English. | 2.07 | 0.96 | -0.09 | -1.735 | 0.083 |
| | 6.There is a demand for study tours, further education abroad, or visiting scholars. | 2.16 | 1.02 | | | |
| Pair 5 | 5. It is necessary to engage in oral communication or presentations related to art profession in English, or to present papers in English at academic conferences. | 2.02 | 0.96 | -0.14 | -2.855 | 0.004** |
| | There is a demand for study tours, further education abroad, or visiting scholars. | 2.16 | 1.02 | | | |

The analysis indicates a differentiated approach to the urgency and importance of various ESP skills, with a general trend favoring interactive and receptive skills over productive ones. The significant emphasis on international experiences and attending lectures suggests that opportunities for direct engagement with global experts and contexts are highly valued, reflecting an understanding of their transformative potential in professional development.

## 5.2 Analysis of students' current ESP competence

Table 4 presents the evaluation of students' ESP competence by the four groups, with the assessment focusing on students' abilities in listening, reading, speaking, writing, and reading strategies in understanding specialized content.

General Difficulties Across ESP Skills: Significant difficulties were reported across all skills, with high agreement percentages indicating challenges in listening (72.87%), speaking (84.09%), reading(85.66%), reading strategies(89.99%) and writing (85.74%). These results suggest a universal need for support in developing these critical ESP skills.

Specific Challenges in Reading and Writing: The highest recorded difficulty was in reading original textbooks and professional literature, as indicated by an average agreement of 89.99%.

**Table 4. Students' current ESP competence.**

| Questions | Respondents | 5 (%) | 4 (%) | 3 (%) | 2 (%) | 1 (%) | Average score | Generally agree (%) | Generally disagree (%) | Non-parametric test | |
|---|---|---|---|---|---|---|---|---|---|---|---|
| | | | | | | | | | | H-value | P-value |
| 7.Students are not accustomed to listening to lectures by foreign experts or teachers, and cannot keep up with the pace and do not know how to take notes. | UF | 26.11 | 34.96 | 30.97 | 6.64 | 1.33 | 3.78 | 61.07 | 7.97 | 18.911 | 0.000** |
| | GS | 38.05 | 40.27 | 19.03 | 2.65 | 0 | 4.14 | 78.32 | 2.65 | | |
| | AT | 42.86 | 35.71 | 17.86 | 3.57 | 0 | 4.18 | 78.57 | 3.57 | | |
| | ET | 41.18 | 32.35 | 20.59 | 2.94 | 2.94 | 4.06 | 73.53 | 5.88 | | |
| | average | 37.05 | 35.82 | 22.11 | 3.95 | 1.07 | 4.04 | 72.87 | 5.02 | | |
| 8.Students read original textbooks and professional literature at a slow speed. | UF | 37.61 | 42.92 | 16.37 | 2.21 | 0.88 | 4.14 | 80.53 | 3.09 | 10.226 | 0.017* |
| | GS | 48.67 | 42.04 | 6.64 | 2.65 | 0 | 4.37 | 90.71 | 2.65 | | |
| | AT | 42.86 | 42.86 | 10.71 | 3.57 | 0 | 4.25 | 85.72 | 3.57 | | |
| | ET | 52.94 | 26.47 | 17.65 | 0.00 | 2.94 | 4.26 | 79.41 | 2.94 | | |
| | average | 45.52 | 38.57 | 12.84 | 2.11 | 0.96 | 4.26 | 84.09 | 3.06 | | |
| 9.Students have difficulties in using English for professional oral communication and participating in academic discussions. | UF | 38.05 | 42.48 | 14.16 | 4.42 | 0.88 | 4.12 | 80.53 | 5.3 | 10.647 | 0.014* |
| | GS | 46.46 | 43.36 | 8.41 | 1.77 | 0 | 4.35 | 89.82 | 1.77 | | |
| | AT | 42.86 | 50 | 3.57 | 0.00 | 3.57 | 4.29 | 92.86 | 3.57 | | |
| | ET | 50 | 29.41 | 14.71 | 0.00 | 5.88 | 4.18 | 79.41 | 5.88 | | |
| | average | 44.34 | 41.31 | 10.21 | 1.55 | 2.58 | 4.24 | 85.66 | 4.13 | | |
| 10.Students have difficulties in writing work description, literature review, abstract, papers, etc. | UF | 36.73 | 43.36 | 15.93 | 3.10 | 0.88 | 4.12 | 80.09 | 3.98 | 15.522 | 0.001** |
| | GS | 48.23 | 44.25 | 6.19 | 1.33 | 0 | 4.39 | 92.48 | 1.33 | | |
| | AT | 50 | 32.14 | 14.29 | 0.00 | 3.57 | 4.25 | 82.14 | 3.57 | | |
| | ET | 55.88 | 32.35 | 0.00 | 8.82 | 2.94 | 4.29 | 88.23 | 11.76 | | |
| | average | 47.71 | 38.03 | 9.10 | 3.31 | 1.85 | 4.26 | 85.74 | 5.16 | | |
| 11.Students lack the methods and necessary knowledge to read original textbooks or works. | UF | 36.28 | 49.56 | 11.5 | 2.21 | 0.44 | 4.19 | 85.84 | 2.65 | 3.524 | 0.318 |
| | GS | 44.25 | 46.46 | 7.08 | 1.33 | 0.88 | 4.32 | 90.71 | 2.21 | | |
| | AT | 57.14 | 32.14 | 7.14 | 0 | 3.57 | 4.39 | 89.28 | 3.57 | | |
| | ET | 64.71 | 29.41 | 2.94 | 0 | 2.94 | 4.53 | 94.12 | 2.94 | | |
| | average | 50.60 | 39.39 | 7.17 | 0.89 | 1.96 | 4.36 | 89.99 | 2.84 | | |

Note: As for this part, teachers are asked to evaluate according to their observation and students are asked to evaluate according to their own experience.

This was closely followed by challenges in writing, with an equal average difficulty rating of 85.74%. Both areas showed the highest demand for enhanced teaching approaches, reflecting significant challenges in both comprehension and production of specialized English content.

Statistical Analysis of Skill Gaps: Significant differences in the assessment of students' skills were evident across different professional contexts and educational levels. For instance, the assessment of listening to professional lectures (Question 7) highlighted a profound disparity (p = 0.000**), suggesting varying perceptions of students' ability to engage with foreign academic content effectively. Slow reading speeds (Question 8) and difficulties in professional oral communication (Question 9) also exhibited significant variances (p = 0.017* and p = 0.014* respectively), pointing to substantial challenges that are recognized differently among the groups.

Writing and Reading Professional Literature: Concerns regarding students' writing abilities and their methods for reading professional literature (Questions 10 and 11) were notably pronounced among art and English teachers. This heightened awareness might reflect the teachers' closer engagement with and higher expectations for students' academic performances in professional settings.

The analysis above underscores a critical need for targeted educational interventions to address the widespread difficulties observed across key ESP skills. The significant disparities in the perception of these difficulties suggest that while students may not fully recognize their skill gaps, teachers are acutely aware of them, particularly in higher-level competencies such as writing and reading specialized texts. This insight calls for a strategic focus on developing curricula that enhance both receptive and productive language skills, with a particular emphasis on reading and writing.

The t-test results in Table 5 indicate significant differences in the evaluation of ESP skills—specifically in listening, reading, speaking, and writing—between undergraduate freshmen and graduate students. The results highlight that graduate students, with their advanced exposure to ESP contexts, tend to have a more acute perception of the challenges associated with ESP skills compared to undergraduates. This awareness is not merely a reflection of increased proficiency but also of an enhanced understanding of the complexities inherent in academic and professional English usage. This differential perception between the groups highlights an essential aspect of ESP education—awareness and acknowledgment of linguistic challenges grow with exposure and experience, suggesting that ESP curricula should be adaptive and responsive to students' evolving educational contexts.

**Table 5. Comparison of assessing students' current ESP competence.**

| Questions | UF-ET comparison | | GS-ET comparison | | AT-ET comparison | | UF-GS comparison | | UF-AT comparison | | GS-AT comparison | |
|---|---|---|---|---|---|---|---|---|---|---|---|---|
| 7.Students are not accustomed to listening to lectures by foreign experts or teachers, and cannot keep up with the pace and do not know how to take notes. | 1.294 | 0.197 | -0.868 | 0.386 | -0.516 | 0.608 | 4.317 | 0.000** | 2.111 | 0.041* | -0.067 | 0.946 |
| 8.Students read original textbooks and professional literature at a slow speed. | -0.069 | 0.945 | -1.3 | 0.201 | -0.456 | 0.65 | 2.614 | 0.009** | 0.517 | 0.606 | -0.724 | 0.47 |
| 9.Students have difficulties in using English for professional oral communication and participating in academic discussions. | -0.135 | 0.892 | -1.446 | 0.156 | -1.259 | 0.213 | 3 | 0.003** | 1.664 | 0.104 | 0.162 | 0.872 |
| 10.Students have difficulties in writing work description, literature review, abstract, papers, etc. | 0.114 | 0.909 | -1.285 | 0.207 | -0.14 | 0.889 | 3.85 | 0.000** | 0.31 | 0.757 | -1.3 | 0.203 |
| 11.Students lack the methods and necessary knowledge to read original textbooks or works. | 1.112 | 0.272 | 0.381 | 0.703 | 0.52 | 0.605 | 1.352 | 0.177 | 0.278 | 0.781 | -0.355 | 0.723 |

## 5.3 Opinions on ESP teaching

Table 6 evaluates the consensus among undergraduate students, graduate students, art teachers, and English teachers on the role of ESP in professional development.

ESP's Role in Professional Competence: There is a high consensus (89.44% agreement) that ESP enhances students' international communication skills and competitiveness (Question12),

**Table 6. Opinions on ESP teaching.**

| Questions | Respondents | 5 (%) | 4 (%) | 3 (%) | 2 (%) | 1 (%) | Average score | Generally agree (%) | Generally disagree (%) | Non-parametric test | |
|---|---|---|---|---|---|---|---|---|---|---|---|
| | | | | | | | | | | H-value | P-value |
| 12. ESP can improve students' international communication and competitiveness in their professional fields. | UF | 40.27 | 47.79 | 11.5 | 0 | 0.44 | 4.27 | 88.06 | 0.44 | 8.133 | 0.043* |
| | GS | 46.46 | 43.81 | 7.08 | 2.21 | 0.44 | 4.34 | 90.27 | 2.65 | | |
| | AT | 60.71 | 39.29 | 0 | 0 | 0 | 4.61 | 100 | 0 | | |
| | ET | 55.88 | 23.53 | 2.94 | 0 | 17.65 | 4 | 79.41 | 17.65 | | |
| | average | 50.83 | 38.61 | 5.38 | 0.55 | 4.63 | 4.31 | 89.44 | 5.19 | | |
| 13. ESP can enhance students' competitiveness in further studies or future employment. | UF | 46.90 | 42.04 | 9.73 | 0.88 | 0.44 | 4.34 | 88.94 | 1.32 | 0.413 | 0.938 |
| | GS | 47.35 | 40.27 | 10.62 | 1.77 | 0 | 4.33 | 87.62 | 1.77 | | |
| | AT | 53.57 | 35.71 | 10.71 | 0 | 0 | 4.43 | 89.28 | 0 | | |
| | ET | 67.65 | 23.53 | 2.94 | 2.94 | 2.94 | 4.5 | 91.18 | 5.88 | | |
| | average | 53.87 | 35.39 | 8.50 | 1.40 | 0.85 | 4.40 | 89.26 | 2.24 | | |
| 14.Teaching ESP is more effective than teaching general English in improving students' language proficiency and skills. | UF | 37.17 | 38.05 | 20.80 | 3.54 | 0.44 | 4.08 | 75.22 | 3.98 | 8.695 | 0.034* |
| | GS | 42.92 | 40.27 | 15.04 | 1.77 | 0 | 4.24 | 83.19 | 1.77 | | |
| | AT | 35.71 | 42.86 | 17.86 | 0 | 3.57 | 4.07 | 78.57 | 3.57 | | |
| | ET | 35.29 | 32.35 | 11.76 | 8.82 | 11.76 | 3.71 | 67.64 | 20.58 | | |
| | average | 37.77 | 38.38 | 16.37 | 3.53 | 3.94 | 4.03 | 76.16 | 7.48 | | |
| 15.Teaching ESP is more effective than teaching general English in motivating students to learn the language. | UF | 34.07 | 36.28 | 26.11 | 3.1 | 0.44 | 4 | 70.35 | 3.54 | 14.27 | 0.003** |
| | GS | 39.82 | 42.92 | 14.60 | 2.65 | 0 | 4.2 | 82.74 | 2.65 | | |
| | AT | 42.86 | 42.86 | 14.29 | 0 | 0 | 4.29 | 85.72 | 0 | | |
| | ET | 41.18 | 23.53 | 23.53 | 11.76 | 0 | 3.94 | 64.71 | 11.76 | | |
| | average | 39.48 | 36.40 | 19.63 | 4.38 | 0.11 | 4.11 | 75.88 | 4.49 | | |
| 16.University students must have a solid foundation in general English, that is, they need to learn general English well before studying ESP. | UF | 36.73 | 42.92 | 15.49 | 4.42 | 0.44 | 4.11 | 79.65 | 4.86 | 1.257 | 0.739 |
| | GS | 40.71 | 42.48 | 12.83 | 3.54 | 0.44 | 4.19 | 83.19 | 3.98 | | |
| | AT | 39.29 | 42.86 | 17.86 | 0 | 0 | 4.21 | 82.15 | 0 | | |
| | ET | 44.12 | 41.18 | 11.76 | 2.94 | 0 | 4.26 | 85.3 | 2.94 | | |
| | average | 40.21 | 42.36 | 14.49 | 2.73 | 0.22 | 4.19 | 82.57 | 2.95 | | |
| 17.As long as university students learn general English well and have a solid foundation, they can meet the requirements of using ESP without studying it specifically. | UF | 21.68 | 30.53 | 33.63 | 13.27 | 0.88 | 3.59 | 52.21 | 14.15 | 3.08 | 0.379 |
| | GS | 26.11 | 22.12 | 30.53 | 19.91 | 1.33 | 3.52 | 48.23 | 21.24 | | |
| | AT | 17.86 | 35.71 | 25 | 14.29 | 7.14 | 3.43 | 53.57 | 21.43 | | |
| | ET | 20.59 | 26.47 | 17.65 | 29.41 | 5.88 | 3.26 | 47.06 | 35.29 | | |
| | average | 21.56 | 28.71 | 26.70 | 19.22 | 3.81 | 3.45 | 50.27 | 23.03 | | |
| 18.College English is a course for general education, and more courses on English culture, literature, and other general knowledge should be offered. | UF | 32.30 | 49.56 | 14.6 | 2.21 | 1.33 | 4.09 | 81.86 | 3.54 | 6.952 | 0.073 |
| | GS | 42.04 | 44.25 | 10.18 | 2.65 | 0.88 | 4.24 | 86.29 | 3.53 | | |
| | AT | 35.71 | 50 | 3.57 | 7.14 | 3.57 | 4.07 | 85.71 | 10.71 | | |
| | ET | 41.18 | 29.41 | 8.82 | 17.65 | 2.94 | 3.88 | 70.59 | 20.59 | | |
| | average | 37.81 | 43.31 | 9.29 | 7.41 | 2.18 | 4.07 | 81.11 | 9.59 | | |
| 19.College English is a practical course, and more ESP courses should be offered. | UF | 32.30 | 49.12 | 15.49 | 1.33 | 1.77 | 4.09 | 81.42 | 3.1 | 5.191 | 0.158 |
| | GS | 40.27 | 43.81 | 12.83 | 2.65 | 0.44 | 4.21 | 84.08 | 3.09 | | |
| | AT | 39.29 | 46.43 | 14.29 | 0.00 | 0.00 | 4.25 | 85.72 | 0 | | |
| | ET | 38.24 | 32.35 | 11.76 | 17.65 | 0.00 | 3.91 | 70.59 | 17.65 | | |
| | average | 37.53 | 42.93 | 13.59 | 5.41 | 0.55 | 4.12 | 80.45 | 5.96 | | |

and that ESP contributes to competitiveness in further studies and future employment (89.26% agreement in Question13), indicating widespread recognition of ESP's career-enhancing benefits.

Comparison of ESP and General English: A significant majority (76.16% agreement, p = 0.034* for Question14) believe ESP is more effective than general English in improving language skills, reflecting a preference for specialized training that directly aligns with professional needs. With 75.88% agreement and a significant difference (p = 0.003**) observed among groups(Question15), respondents recognize ESP as a more motivating approach than general English. This suggests that the relevance of ESP to students' professional interests may enhance their engagement and learning outcomes.

Necessity of a General English Foundation: Despite the focus on specialized training, 82.57% of respondents(Question 16) agree that a solid general English foundation is necessary before studying ESP. This highlights an understanding of the importance of core language skills as a basis for specialized learning.

Adequacy of General English for Professional Needs: Only 50.27% believe that general English can meet professional communication needs(Question 17), indicating that many see general English as insufficient for specialized contexts. This lower agreement underscores the necessity for ESP courses that address specific professional language requirements.

Curriculum Recommendations: There is a strong consensus on the need for a balanced curriculum that includes both cultural and practical language skills (81.11% and 80.45% agreement respectively for questions 18 and 19). However, there are significant differences among the groups (p-values near 0.05 for both questions), suggesting diverse opinions on how much emphasis to place on cultural versus practical aspects of English teaching.

The data in Table 6 reflects a strong endorsement of ESP's role in enhancing professional skills and international competitiveness. However, the significant differences among respondent groups in their evaluations of ESP's effectiveness in various aspects suggest that experiences and expectations can influence perceptions of ESP's value. The recognition of the need for a general English foundation, coupled with the call for more specialized courses, indicates a comprehensive approach to English education that addresses both foundational skills and specific professional requirements.

Table 7 provides a detailed t-test analysis evaluating differences in perceptions among undergraduate students, graduate students, art teachers, and English teachers regarding the effectiveness of ESP. The analysis highlights the varying degrees of recognition and agreement about the role of ESP in professional and academic contexts.

ESP's Role in Enhancing International Competence: Art teachers show a significantly higher level of recognition of ESP's role in enhancing students' international communication and competitiveness compared to English teachers, graduate students and undergraduate freshmen (p = 0.007**, p < 0.001** for Question12). This could suggest that art teachers, often engaged in more globally connected disciplines, recognize the critical value of specialized English in international settings.

Effectiveness of ESP vs. General English: There is a significant difference in perceptions regarding the effectiveness of ESP over general English in enhancing language proficiency. Graduate students, with more direct experience in ESP, affirm its benefits more strongly than undergraduate freshmen(0.022* for Question14) and English teachers(p = 0.023*), who may prioritize foundational English skills. The motivation aspect shows significant differences, with graduate students and art teachers finding ESP more motivating than English teachers and undergraduate freshmen do (Question 15). This suggests that the relevance of ESP to professional and creative fields may increase student engagement and enthusiasm for learning.

**Table 7. Comparison of opinions on ESP teaching.**

| Questions | UF-ET comparison | | GS-ET comparison | | AT-ET comparison | | UF-GS comparison | | UF-AT comparison | | GS-AT comparison | |
|---|---|---|---|---|---|---|---|---|---|---|---|---|
| | t | p | t | p | t | p | t | p | t | p | t | p |
| ESP can improve students' international communication and competitiveness in their professional fields. | -1.911 | 0.064 | -1.897 | 0.066 | -2.862 | 0.007** | -0.016 | 0.988 | 5.422 | 0.000** | 4.622 | 0.000** |
| 13. ESP can enhance students' competitiveness in further studies or future employment. | -0.333 | 0.739 | -0.072 | 0.943 | -0.366 | 0.715 | -0.507 | 0.612 | 0.223 | 0.824 | 0.442 | 0.659 |
| 14.Teaching ESP is more effective than teaching general English in improving students' language proficiency and skills. | -1.636 | 0.11 | -2.379 | 0.023* | -1.624 | 0.11 | 2.293 | 0.022* | 0.381 | 0.703 | -0.723 | 0.47 |
| 15.Teaching ESP is more effective than teaching general English in motivating students to learn the language. | -1.074 | 0.289 | -2.172 | 0.036* | -2.364 | 0.022* | 2.866 | 0.004** | 2.513 | 0.016* | 0.621 | 0.535 |
| 16.University students must have a solid foundation in general English, that is, they need to learn general English well before studying ESP. | 0.772 | 0.441 | 0.349 | 0.728 | 0.019 | 0.985 | 0.89 | 0.374 | 0.694 | 0.488 | 0.302 | 0.763 |
| 17.As long as university students learn general English well and have a solid foundation, they can meet the requirements of using ESP without studying it specifically. | -1.567 | 0.125 | -1.026 | 0.306 | -0.916 | 0.364 | -1.466 | 0.143 | -0.359 | 0.72 | 0.324 | 0.746 |
| 18. College English is a course for general education, and more courses on English culture, literature, and other general knowledge should be offered. | -1.955 | 0.058 | -2.259 | 0.030* | -1.338 | 0.186 | 0.965 | 0.335 | -0.335 | 0.738 | -0.795 | 0.427 |
| 19. College English is a practical course, and more ESP courses should be offered. | -1.835 | 0.075 | -2.023 | 0.05 | -2.171 | 0.035* | 0.575 | 0.566 | 0.776 | 0.439 | 0.52 | 0.604 |

Curricular Integration of ESP: English teachers show significantly lower agreement rates regarding the inclusion of more specialized English courses alongside general education (p = 0.030* and p = 0.035* for Questions 18 and 19, respectively). This difference indicates a more conservative or traditional view among English teachers about integrating specialized English into the curriculum.

The analysis in Table 7 underscores significant inter-group differences in perceptions of ESP's role. Art teachers and graduate students, who are likely more exposed to the direct applications of specialized English, recognize its benefits more distinctly than English teachers. This divergence in views suggests varying experiences with and expectations from specialized English training across different educational roles.

## 5.4 Teaching situation and challenges

The non-parametric tests conducted in Tables 8 and 9 provide insight into the commonalities and differences in perceptions regarding the challenges encountered in ESP education, indicating systemic issues in ESP education.

Faculty Qualifications (Question 20): A notable consensus exists that the lack of qualified teachers is a major challenge in ESP teaching, with an average agreement rate of 73.49%. However, art teachers particularly highlight this issue, with a significantly higher agreement rate of 89.28% and significant differences with the other three groups(p = 0.035*, 0.012*, 0.020* in Table 9), suggesting that within their discipline, the quality of teaching staff is perceived as a critical barrier to effective ESP education and this is possibly due to the specific demands of integrating art and language learning which require highly specialized skills.

Textbook Quality (Question 21): Concerns about the adequacy of ESP textbooks are widespread, with an average agreement rate of 76.03%. English teachers, in particular, express a higher level of concern (85.3% agreement), indicating a strong belief that the quality and relevance of educational materials are pivotal for ESP teaching effectiveness.

**Table 8. Teaching situation and challenges.**

| Questions | Respondents | 5 (%) | 4 (%) | 3 (%) | 2 (%) | 1 (%) | Average | Generally agree (%) | Generally disagree (%) | Non-parametric test | |
|---|---|---|---|---|---|---|---|---|---|---|---|
| | | | | | | | | | | H-value | P-value |
| 20.The lack of qualified teachers is one of the main problems in conducting specialized English teaching. | UF | 26.55 | 40.71 | 26.99 | 5.31 | 0.44 | 3.88 | 67.26 | 5.75 | 5.621 | 0.132 |
| | GS | 27.88 | 38.94 | 29.65 | 3.54 | 0 | 3.91 | 66.82 | 3.54 | | |
| | AT | 28.57 | 60.71 | 7.14 | 3.57 | 0 | 4.14 | 89.28 | 3.57 | | |
| | ET | 29.41 | 41.18 | 8.82 | 5.88 | 14.71 | 3.65 | 70.59 | 20.59 | | |
| | average | 28.10 | 45.39 | 18.15 | 4.58 | 3.79 | 3.90 | 73.49 | 8.36 | | |
| 21.The lack of good specialized English textbooks is one of the main problems in conducting specialized English teaching. | UF | 23.89 | 47.79 | 22.57 | 5.31 | 0.44 | 3.89 | 71.68 | 5.75 | 2.616 | 0.455 |
| | GS | 30.97 | 41.15 | 25.22 | 2.21 | 0.44 | 4 | 72.12 | 2.65 | | |
| | AT | 21.43 | 53.57 | 17.86 | 7.14 | 0 | 3.89 | 75 | 7.14 | | |
| | ET | 47.06 | 38.24 | 8.82 | 2.94 | 2.94 | 4.24 | 85.3 | 5.88 | | |
| | average | 30.84 | 45.19 | 18.62 | 4.40 | 0.96 | 4.01 | 76.03 | 5.36 | | |
| 22.The inability to change mindset is one of the main problems in conducting specialized English teaching. | UF | 26.55 | 46.90 | 23.45 | 3.10 | 0 | 3.97 | 73.45 | 3.1 | 2.216 | 0.529 |
| | GS | 29.65 | 47.79 | 21.24 | 1.33 | 0 | 4.06 | 77.44 | 1.33 | | |
| | AT | 21.43 | 46.43 | 28.57 | 3.57 | 0 | 3.86 | 67.86 | 3.57 | | |
| | ET | 41.18 | 38.24 | 11.76 | 5.88 | 2.94 | 4.09 | 79.42 | 8.82 | | |
| | average | 29.70 | 44.84 | 21.26 | 3.47 | 0.74 | 4.00 | 74.54 | 4.21 | | |
| 23. The low English proficiency of students is one of the main problems in conducting specialized English teaching. | UF | 28.76 | 45.58 | 20.35 | 4.42 | 0.88 | 3.97 | 74.34 | 5.3 | 6.251 | 0.1 |
| | GS | 33.63 | 46.9 | 16.37 | 3.1 | 0 | 4.11 | 80.53 | 3.1 | | |
| | AT | 32.14 | 32.14 | 17.86 | 10.71 | 7.14 | 3.71 | 64.28 | 17.85 | | |
| | ET | 44.12 | 35.29 | 2.94 | 8.82 | 8.82 | 3.97 | 79.41 | 17.64 | | |
| | average | 34.66 | 39.98 | 14.38 | 6.76 | 4.21 | 3.94 | 74.64 | 10.97 | | |

Conceptual Mindset (Question 22): The challenge of changing educational mindsets towards ESP teaching is recognized, with an average agreement rate of 74.54%. English teachers again report higher concern (79.42% agreement), which may reflect difficulties in integrating innovative ESP approaches within traditional language teaching frameworks.

Student Proficiency (Question 23): The low English proficiency of students is acknowledged as a significant impediment to ESP teaching, with an overall agreement rate of 74.64%. Art teachers and English teachers indicate substantial concern about this issue, with lower agreement rates reflecting perhaps greater expectations or observations of insufficient language skills impacting specialized learning.

**Table 9. Comparative analysis of the perceptions towards teaching situation and challenges.**

| Questions | UF-ET comparison | | GS-ET comparison | | AT-ET comparison | | UF-GS comparison | | UF-AT comparison | | GS-AT comparison | |
|---|---|---|---|---|---|---|---|---|---|---|---|---|
| | t | p | t | p | t | p | t | p | t | p | t | p |
| 20.The lack of qualified teachers is one of the main problems in conducting specialized English teaching. | -0.764 | 0.449 | -0.907 | 0.37 | -2.164 | 0.035* | 0.379 | 0.705 | 2.619 | 0.012* | 2.43 | 0.020* |
| 21.The lack of good specialized English textbooks is one of the main problems in conducting specialized English teaching. | 1.376 | 0.176 | 1.041 | 0.299 | 0.791 | 0.432 | 0.739 | 0.46 | 0.189 | 0.851 | -0.153 | 0.879 |
| The inability to change mindset is one of the main problems in conducting specialized English teaching. | 0.055 | 0.956 | -0.621 | 0.535 | 0.413 | 0.681 | 1.314 | 0.189 | -0.546 | 0.585 | -1.076 | 0.29 |
| 23.The low English proficiency of students is one of the main problems in conducting specialized English teaching. | -0.469 | 0.642 | -1.14 | 0.262 | 0.765 | 0.447 | 1.824 | 0.069 | -1.413 | 0.168 | -2.023 | 0.052 |

The data from Tables 8 and 9 indicates a shared perception among all groups that teacher qualifications, textbook quality, resistance to new educational paradigms, and student language proficiency are significant barriers to effective ESP teaching. This widespread concern implies a need for systemic reforms in ESP teacher training, curriculum development, textbook quality, and student language preparation. The particularly high concern among art teachers about faculty qualifications could suggest a demand for more specialized training within their fields, likely due to the specific language requirements of art-related disciplines. The heightened sensitivity of English teachers to textbook quality and mindset changes suggests they may see these factors as directly impacting the adaptability and success of ESP programs. Their focus on foundational language skills might also explain their concern over student proficiency as a primary challenge.

Table 10 employs paired t-tests to analyze the relative severity of various challenges encountered in ESP teaching. In the analysis presented, teacher issues are highlighted as significantly more critical than other challenges in ESP teaching when compared to textbook issues, student proficiency issues, and conceptual issues, demonstrating statistically significant differences with respective p-values of 0.033*, 0.011*, and 0.001**. This finding suggests that deficiencies in teacher training and qualifications may directly hinder the delivery of quality ESP instruction more so than limitations in textbooks, student proficiency, or educational mindsets. The lack of significant differences among textbook quality, conceptual challenges, and student proficiency implies a perceived uniformity in their impact on teaching effectiveness. This suggests that while important, improvements in these areas alone may not suffice without addressing the more critical teacher issues.

**Table 10. Comparative analysis of the challenges in implementing ESP teaching.**

| Pairs | Questions | Average | Standard deviation | Mean difference | T-value | P-value |
|---|---|---|---|---|---|---|
| Pair 6 | 20.The lack of qualified teachers is one of the main problems in conducting specialized English teaching. | 2.37 | 0.59 | 0.06 | 2.14 | 0.033* |
| | 21.The lack of good specialized English textbooks is one of the main problems in conducting specialized English teaching. | 2.32 | 0.55 | | | |
| Pair 7 | 20.The lack of qualified teachers is one of the main problems in conducting specialized English teaching. | 2.37 | 0.59 | 0.1 | 3.23 | 0.001** |
| | 22.The inability to change mindset is one of the main problems in conducting specialized English teaching. | 2.28 | 0.5 | | | |
| Pair 8 | 20.The lack of qualified teachers is one of the main problems in conducting specialized English teaching. | 2.37 | 0.59 | 0.08 | 2.542 | 0.011* |
| | 23.The low English proficiency of students is one of the main problems in conducting specialized English teaching. | 2.29 | 0.57 | | | |
| Pair 9 | 21.The lack of good specialized English textbooks is one of the main problems in conducting specialized English teaching. | 2.32 | 0.55 | 0.04 | 1.668 | 0.096 |
| | 22.The inability to change mindset is one of the main problems in conducting specialized English teaching. | 2.28 | 0.5 | | | |
| Pair10 | 21.The lack of good specialized English textbooks is one of the main problems in conducting specialized English teaching. | 2.32 | 0.55 | 0.02 | 0.802 | 0.423 |
| | 23.The low English proficiency of students is one of the main problems in conducting specialized English teaching. | 2.29 | 0.57 | | | |
| Pair 11 | 22.The inability to change mindset is one of the main problems in conducting specialized English teaching. | 2.28 | 0.5 | -0.02 | -0.669 | 0.504 |
| | 23.The low English proficiency of students is one of the main problems in conducting specialized English teaching. | 2.29 | 0.57 | | | |

## 6. Discussion

The investigation into the needs analysis of ESP courses in colleges of art and design has elucidated a multifaceted view of the consensus and divergence among key educational stakeholders—undergraduate freshmen, graduate students, art teachers, and English teachers. The results depict a nuanced landscape of needs, perceptions, and challenges that influence ESP teaching and learning in these specialized fields.

### 6.1 Consensus on the role and challenges of ESP

The study underscores a strong consensus across all respondent groups about the pivotal role of ESP in bolstering students' international competitiveness and enhancing their professional capabilities. The high agreement rates documented suggest a universal recognition that ESP is superior to general English in not only improving language proficiency but also in engaging students more deeply in their learning processes. This consensus supports the integration of specialized English into curricula to better prepare students for global professional demands, affirming findings in the literature that emphasize the need for context-specific language instruction [36–38].

While there is a strong general consensus on the value of ESP in enhancing professional capabilities and competitiveness, there is a widespread acknowledgement of students' difficulties across ESP skills, indicating a gap between student competencies and the professional demands of the art and design fields. This study highlights the critical need for targeted educational interventions to effectively address the difficulties identified.

Moreover, there is an unanimous recognition of the significant challenges encountered in teaching ESP. These challenges span from faculty qualifications and teaching materials to student proficiency and conceptual understanding. Critically, teacher-related issues emerged as the most pronounced barriers, as evidenced by significant differences in perceptions particularly noted among art teachers. This finding is consistent with scholarly discussions that stress the importance of qualified and well-prepared instructors in the effectiveness of language teaching [39]. This likely explains the significant increase in research on teachers' professional development over the past decade, which mirrors practitioners' aspirations to enhance both their classroom competencies and their academic identities [40].

### 6.2 Divergences in perceptions

Despite widespread agreement on several fronts, the analysis also revealed notable divergences in perceptions, particularly between art teachers and other educational groups. Art teachers uniquely emphasized faculty issues more than their counterparts, likely reflecting specific disciplinary needs where effective ESP teaching requires not only language proficiency but also a deep understanding of artistic concepts and terminology. This perspective is indicative of the specialized demands of art-related disciplines that may not be as pronounced in more general educational settings. Addressing this issue requires targeted professional development programs that equip teachers with the necessary skills to bridge the gap between language and content in art and design education.

Conversely, English teachers were particularly concerned with the quality of teaching materials, indicating a belief that the effectiveness of ESP teaching is heavily reliant on the availability of high-quality, relevant educational resources. This finding resonates with evidence from ESP research indicating that ESP teachers require substantial support in terms of materials and professional development [37]. This divergence underscores the varying emphases placed on different aspects of ESP program delivery, which may influence the prioritization of resources and training in different departments.

The role of general English as a foundation for specialized English also highlighted divergent views. While there was a general consensus on the necessity of a solid general English background, views on its sufficiency for professional communication were mixed, echoing debates in ESP education regarding the balance between general and specialized language training [41]. The varied perspectives on this issue suggest that while foundational English skills are critical, they may not fully meet the specific linguistic demands of professional fields, thus necessitating tailored ESP courses.

### 6.3 Implications and future directions

The findings from this study suggest several implications for curriculum development and instructional strategies in ESP courses. First, there is a clear need for enhanced focus on teacher training and development to address the identified challenges related to faculty qualifications. Second, the development and curation of specialized teaching materials should be prioritized to support effective instruction across various disciplines within art and design colleges.

Future research should explore the specific training needs of ESP instructors within the context of art and design to better tailor professional development programs. Some attempts have already been made to enhance the professional development of English-medium instruction teachers [42]. Additionally, studies could investigate the impact of enhanced teaching materials and innovative instructional strategies on student outcomes in ESP courses, further contributing to the refinement of ESP pedagogy in specialized educational settings.

## 7. Conclusion and implications for ESP course design

This study provides a nuanced analysis of the needs and challenges associated with English for Specific Purposes(ESP) courses within art and design colleges, revealing key insights into both consensus and divergence among various educational stakeholders, including undergraduate students, graduate students, art teachers, and English teachers. Through detailed examination, a distinct demand has been identified among students for specialized English learning, especially in interactive and receptive skills. This demand correlates with the urgent need to enhance reading abilities as recognized by both students and teachers. The noted discrepancy between teacher evaluations of student abilities and students' self-assessments, particularly among undergraduates, underscores a significant proficiency gap that must be addressed to meet professional requirements effectively.

Despite widespread acknowledgment of ESP's positive impact on enhancing professional skills, the enthusiasm for adopting ESP courses varies, with notable resistance from some English teachers. This reluctance to transition from English for General Purposes (EGP) to ESP, despite the evident demand from both art teachers and students, presents a substantial barrier to the successful integration of ESP into curricula.

A recurring theme from this study is the pivotal role of teacher-related issues as the primary obstacle in ESP course implementation. Teachers particularly highlight challenges with the quality of textbooks, suggesting the necessity for a holistic approach to ESP course development that encompasses both content quality and effective delivery methods, including enhancing teacher readiness and improving resource availability.

The difficulties related to teacher qualifications and the quality of instructional materials call for strategic interventions. These should include targeted faculty development programs and the creation of context-specific teaching resources. However, research on ESP teachers remains scant [17], indicating a gap that needs filling. One promising strategy could involve a collaborative approach where English teachers at art colleges work closely with art faculty and industry professionals to develop specialized materials that meld disciplinary knowledge with

linguistic and educational expertise [42–44]. Such partnerships could foster ESP courses that are not only linguistically robust but also deeply integrated with art and design professional contexts.

Moreover, the findings emphasize the necessity for a balanced educational strategy that encompasses both general English and specialized English training. This approach ensures that students are equipped not only with foundational language skills but also with the specialized language skills necessary for their professional development.

The divergences in perception among different educational groups highlight the need for a nuanced approach to ESP course design. It is evident that universal solutions may not suffice; instead, tailored courses designed to meet the specific requirements of various stakeholders, particularly in specialized fields like art and design, are essential.

To address the identified needs and challenges associated with ESP courses in art and design colleges, several pedagogical strategies can be implemented.

1. Targeted Faculty Development Programs: Comprehensive faculty development programs are essential for equipping teachers with the skills needed to deliver effective ESP instruction. These programs should focus on the unique linguistic requirements of art and design contexts, integrating both disciplinary knowledge and pedagogical techniques. By enhancing teachers' understanding of art-specific language and instructional strategies, these programs can bridge the gap between general English proficiency and specialized professional language needs.

2. Collaborative Material Development: Developing high-quality, context-specific teaching materials requires collaboration between English teachers, art faculty, and industry professionals. This collaborative approach ensures that learning resources are relevant, practical, and aligned with the specific needs of students in art and design fields. Materials should meld disciplinary knowledge with linguistic and educational expertise, providing students with robust language skills that are deeply integrated with their professional contexts.

3. Integrated Curriculum Design: An integrated curriculum that balances general English proficiency with specialized ESP content is crucial for student success. Such a curriculum should ensure that students build a strong foundation in general English while progressively acquiring the specialized vocabulary and skills necessary for their professional development.

4. Enhanced Teacher Readiness: Improving teacher readiness is vital for the successful implementation of ESP courses. Ongoing professional development opportunities, including workshops, seminars, and access to the latest research in ESP methodologies, are necessary. These initiatives should focus on equipping teachers with the skills and knowledge required to deliver high-quality ESP instruction, addressing the challenges identified in the study.

5. Resource Availability: Increasing the availability of high-quality instructional resources that are tailored to the needs of art and design students, including textbooks, digital tools, and multimedia content, is essential for effective ESP instruction.

6. Feedback Mechanisms: Establishing robust feedback mechanisms that regularly gather input from students and teachers on the effectiveness of ESP courses is crucial. This feedback should be used to continuously refine and improve the curriculum and instructional methods, ensuring that the courses remain relevant and effective in meeting the needs of students and educators.

7. Institutional Support: Strong institutional support for ESP initiatives is necessary to ensure their successful implementation. This includes administrative backing and adequate funding

for program development, teacher training, and resource acquisition. Institutional commitment to ESP can facilitate the integration of specialized language instruction into the broader curriculum, enhancing the overall quality of education in art and design colleges.

8. Global Perspectives: Considering global perspectives and practices in ESP by benchmarking against international standards and incorporating best practices from other institutions worldwide can broaden the applicability of the research findings. This approach addresses the limitations of context-specific findings and ensures that ESP courses are aligned with global educational trends and standards.

In conclusion, the research clearly articulates the need for dedicated and specifically tailored ESP courses in art and design colleges. To effectively meet these identified needs, a concerted effort from educators, administrators, and field professionals is imperative. By addressing the distinct challenges and leveraging collaborative strategies proposed herein, ESP courses can be optimally designed to significantly enhance students' international competitiveness and professional prowess in the art and design sector.

This study, while comprehensive in its exploration of the needs and challenges associated with ESP courses in art and design colleges, is subject to limitations that must be considered when interpreting the findings. The data for this research were collected from only two colleges of art and design in China. This limitation restricts the generalizability of the findings to other institutions with different educational contexts and cultures. Educational practices and the perceived importance of ESP can vary significantly across different regions and educational systems. Therefore, the insights gained may not fully represent the broader spectrum of challenges and needs faced by art and design colleges globally.

## Supporting information

**S1 File. Questionnaire for ESP needs survey.**
(DOC)

**S1 Table. Original collected data.**
(XLS)

## Acknowledgments

We would like to thank all the teachers and students who participated in the survey.

## Author Contributions

**Conceptualization:** Fengfan Mao.

**Data curation:** Fengfan Mao, Jiefeng Zhou.

**Formal analysis:** Fengfan Mao, Jiefeng Zhou.

**Funding acquisition:** Fengfan Mao.

**Investigation:** Fengfan Mao, Jiefeng Zhou.

**Methodology:** Fengfan Mao.

**Project administration:** Fengfan Mao.

**Writing – original draft:** Fengfan Mao.

**Writing – review & editing:** Jiefeng Zhou.

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
