## [Decision Letter · Decision Letter 0]

6 Dec 2023

PONE-D-23-32993A Needs Analysis of ESP Courses in Colleges of Art and Design: Consensus and DivergencePLOS ONE

Dear Dr. Mao,

Thank you for submitting your manuscript to PLOS ONE. After careful consideration, we feel that it has merit but does not fully meet PLOS ONE’s publication criteria as it currently stands. Therefore, we invite you to submit a revised version of the manuscript that addresses the points raised during the review process.

                  The reviewer's comments (Major Revisions):The study reports a needs analysis study with teachers and students. The main comments go to:1. The literature review part is mainly descriptive. More critical comments are expected.2. There is no theoretical framework.3. The data analysis is mainly descriptive, whereas it needs to be inferential.4. A critical discussion is missing.5. The pedagogical implications are needed in the conclusion.Please ensure that your decision is justified on PLOS ONE’s publication criteria and not, for example, on novelty or perceived impact.

We look forward to receiving your revised manuscript.

Kind regards,

Ali Sorayyaei Azar, PhD

Academic Editor

PLOS ONE

Journal Requirements:

Additional Editor Comments:

Dear Respectful Author(s),

Greetings.

The study reports a needs analysis study with teachers and students. The main comments go to:

1. The literature review part is mainly descriptive. More critical comments are expected.

2. There is no theoretical framework.

3. The data analysis is mainly descriptive, whereas it needs to be inferential.

4. A critical discussion is missing.

5. The pedagogical implications are needed in the conclusion.

Your paper needs to be revised and refined according to the comments posted above.

Thank you!

Reviewers' comments:

Reviewer's Responses to Questions

**Comments to the Author**

1. Is the manuscript technically sound, and do the data support the conclusions?

Reviewer #1: Yes

2. Has the statistical analysis been performed appropriately and rigorously? 

Reviewer #1: No

3. Have the authors made all data underlying the findings in their manuscript fully available?

Reviewer #1: No

4. Is the manuscript presented in an intelligible fashion and written in standard English?

Reviewer #1: Yes

5. Review Comments to the Author

Reviewer #1: The study reports a needs analysis study with teachers and students. The literature review part is mainly descriptive. More critical comments are expected. There is no theoretical framework. The data analysis is mainly also descriptive. A critical discussion is missing. The pedagogical implications are needed in the conclusion.

6. PLOS authors have the option to publish the peer review history of their article (what does this mean?). If published, this will include your full peer review and any attached files.

Reviewer #1: No

---

## [Author Response · Author response to Decision Letter 0]

11 Jan 2024

Dear Dr,

I would like to express my gratitude to you for the insightful comments and suggestions. I have carefully considered each point raised and have made significant revisions to the manuscript accordingly. Below, I detail how each comment was addressed:

Reviewer's Comment: Descriptive Literature Review 

Revision Made: I have enriched the Literature Review section by incorporating more critical commentaries, thereby enhancing its critical depth and academic rigor. This addition aims to provide a more analytical perspective on existing literature.

Reviewer's Comment: Lack of Theoretical Framework 

Revision Made: A theoretical framework has been developed and included, based on the insights gained from the revised literature review and the research design. This framework provides a stronger academic foundation for the study.

Reviewer's Comment: Descriptive Data Analysis 

Revision Made: The data analysis section has been thoroughly revised. The focus now is on interpreting the implications and potential reasons behind the observed statistical trends and differences, moving beyond mere description to more inferential analysis.

Reviewer's Comment: Missing Critical Discussion 

Revision Made: A comprehensive discussion section has been added. This section ties directly to the study’s title and is based on the updated data analysis, offering a critical examination of the findings in the context of existing literature.

Reviewer's Comment: Absence of Pedagogical Implications in Conclusion 

Revision Made: The Conclusion and Implications section has been rewritten to explicitly include pedagogical implications. This addition is intended to highlight the practical relevance of the study’s findings in educational contexts.

Additionally, I have made the following improvements:

Introduction: The introduction has been revised for enhanced clarity and coherence, setting a clear context for the study.

Supporting Information: The questionnaire used in the research, along with the original collected data, has been included in the Supporting Information part, providing a comprehensive view of the research methodology.

I believe these revisions address the concerns raised by the reviewers and enhance the manuscript's contribution to the field. I hope that the revised manuscript is now suitable for publication in PLOS ONE.

Thank you for considering my submission, and I look forward to hearing from you.

Sincerely,

Fengfan Mao

Hubei Institute of Fine Arts

---

## [Editor Report · Decision Letter 1]

17 Jan 2024

PONE-D-23-32993R1A needs analysis of ESP courses in colleges of art and design: Consensus and divergencePLOS ONE

Dear Dr. Mao,

Thank you for submitting your manuscript to PLOS ONE. After careful consideration, we feel that it has merit but does not fully meet PLOS ONE’s publication criteria as it currently stands. Therefore, we invite you to submit a revised version of the manuscript that addresses the points raised during the review process.

**You are required to insert all the comments, revise the paper with highlighted areas, and resubmit it to the portal. Please follow up with the revision instructions. I could not see 'Revised Manuscript with Track Changes' and a separate file labeled 'Manuscript'.**

We look forward to receiving your revised manuscript.

Kind regards,

Ali Sorayyaei Azar, PhD

Academic Editor

PLOS ONE

Journal Requirements:

Additional Editor Comments:

Dear Respective author(s),

Greetings! I hope this message finds you well.

You are required to include all the comments in your revised paper and resubmit the revised and refined paper to us. All the revised sections should be highlighted to indicate that the sections have already been revised. This is the protocol at PLOS ONE.

Please take the needed action.

Thank you!

Regards,

Dr Ali S Azar

---

## [Author Response · Author response to Decision Letter 1]

18 Jan 2024

As you couldn't see the Revised Manuscript with Track Changes and Manuscript(revised clean version), I have resubmitted them as separate files to the portal.

---

## [Editor Report · Decision Letter 2]

19 Apr 2024

PONE-D-23-32993R2A needs analysis of ESP courses in colleges of art and design: Consensus and divergencePLOS ONE

Dear Dr. Mao,

Thank you for submitting your manuscript to PLOS ONE. After careful consideration, we feel that it has merit but does not fully meet PLOS ONE’s publication criteria as it currently stands. Therefore, we invite you to submit a revised version of the manuscript that addresses the points raised during the review process.

We look forward to receiving your revised manuscript.

Kind regards,

Ali Derakhshan

Academic Editor

PLOS ONE

Journal Requirements:

**Additional Editor Comments:**

Dear Dr. Mao,

Please revise and highlight your paper based on the previous comments.

Best,

Ali Derakhshan

---

## [Author Response · Author response to Decision Letter 2]

23 May 2024

Dear Dr. Derakhshan, 

Thank you for your valuable feedback on my manuscript titled “A Needs Analysis of ESP Courses in Colleges of Art and Design: Consensus and Divergence”. I have carefully considered your comments and made substantial revisions to improve the quality of the paper. Below is a summary of the changes made in response to your suggestions:

Literature Review: I have restructured and rewritten the literature review section with a new structure and perspective. The revised version includes more critical comments and a thorough analysis of the existing literature to highlight the gaps and the relevance of my study.

Theoretical Framework: Based on the revised literature review, I have enriched the theoretical framework. This section now clearly outlines the theories and models that underpin the research.

Data Analysis: The data analysis section has been reformulated from a new perspective to include inferential statistics. I have incorporated a more robust analysis of the data, moving beyond descriptive statistics to offer deeper insights. Moreover, the formatting of tables in this part has been enhanced.

Discussion: The discussion section has been revised to reflect the new data analysis. It now includes a critical discussion of the findings, their implications, and how they relate to the existing literature.

Pedagogical Implications: In the conclusion section, I have added the pedagogical implications of my findings. This part highlights how the results can be applied in educational settings and suggests practical recommendations for educators.

Additionally, I have made the following improvements:

Abstract: The abstract part has been improved to respond to the changes in other parts of the paper.

References：As the paper undergoes revisions, additional references are included in the reference list. 

I believe these revisions have significantly strengthened the manuscript, and I look forward to your feedback.

Thank you for your guidance and consideration.

Best regards,

Fengfan Mao

Hubei Institute of Fine Arts

---

## [Editor Report · Decision Letter 3]

27 May 2024

A needs analysis of ESP courses in colleges of art and design: Consensus and divergence

PONE-D-23-32993R3

Dear Dr. Mao,

We’re pleased to inform you that your manuscript has been judged scientifically suitable for publication and will be formally accepted for publication once it meets all outstanding technical requirements.

Kind regards,

Ali Derakhshan

Academic Editor

PLOS ONE

Additional Editor Comments (optional):

Dear Dr Mao,

Thank you for your submission.

Best,

Ali
---

## [Editor Report · Acceptance letter]

31 May 2024

PONE-D-23-32993R3 

PLOS ONE

Dear Dr. Mao, 

I'm pleased to inform you that your manuscript has been deemed suitable for publication in PLOS ONE. Congratulations! Your manuscript is now being handed over to our production team.

Kind regards, 

on behalf of

Dr. Ali Derakhshan 

Academic Editor

PLOS ONE